# Phosphoproteomic analysis of lettuce (*Lactuca sativa* L.) reveals starch and sucrose metabolism functions during bolting induced by high temperature

**Xiaoxiao Qin**[1☯], **Panpan Li**[1☯], **Shaowei Lu**[2], **Yanchuan Sun**[1], **Lifeng Meng**[3], **Jinghong Hao**[1]*, **Shuangxi Fan**[1]*

1 Beijing Key Laboratory of New Technology in Agricultural Application, National Demonstration Center for Experimental Plant Production Education, Plant Science and Technology College, Beijing University of Agriculture, Beijing, China, 2 Facility Horticulture Institute, Ministry of Agriculture Planning and Design Research Academy, Beijing, China, 3 Institute of Apicultural Research/Key Laboratory of Pollinating Insect Biology, Ministry of Agriculture, Chinese Academy of Agricultural Science, Beijing, China

☯ These authors contributed equally to this work.
* haojinghong2013@126.com (JH); fsx20@bua.edu.cn (SF)

**Data Availability Statement:** All relevant data are within the paper and its Supporting information files.

## Abstract

High temperatures induce early bolting in lettuce (Lactuca sativa L.), which decreases both quality and production. However, knowledge of the molecular mechanism underlying high temperature promotes premature bolting is lacking. In this study, we compared lettuce during the bolting period induced by high temperatures (33/25 ˚C, day/night) to which raised under controlled temperatures (20/13 ˚C, day/night) using iTRAQ-based phosphoproteomic analysis. A total of 3,814 phosphorylation sites located on 1,766 phosphopeptides from 987 phosphoproteins were identified after high-temperature treatment, among which 217 phosphoproteins significantly changed their expression abundance (116 upregulated and 101 downregulated). Most phosphoproteins for which the abundance was altered were associated with the metabolic process, with the main molecular functions were catalytic activity and transporter activity. Regarding the functional pathway, starch and sucrose metabolism was the mainly enriched signaling pathways. Hence, high temperature influenced phosphoprotein activity, especially that associated with starch and sucrose metabolism. We suspected that the lettuce shorten its growth cycle and reduce vegetative growth owing to changes in the contents of starch and soluble sugar after high temperature stress, which then led to early bolting/flowering. These findings improve our understanding of the regulatory molecular mechanisms involved in lettuce bolting.

## Introduction

Plants can stimulate procreation under abiotic stress conditions, and gibberellins play an important role in lettuce bolting [1]. Bolting and flowering times are crucial to ensure

**Funding:** This study was supported by the National Natural Science Foundation of China (Grant No. 32072560); the 2018 Joint Funding Project of Beijing Natural Science Foundation-the Municipal Education Commission (KZ201810020027); Beijing Leafy Vegetables Innovation Team of Modern Agro-industry Technology Research System (BAIC07-2021); The Construction of Beijing Science and Technology Innovation and Service Capacity in Top Subjects (CEFF-PXM2019_014207_000032). Beijing University of Agriculture-Three Funds-Youth Humanities Foundation (No.5077516001/003).

**Competing interests:** The authors have declared that no competing interests exist.

reproductive success and high agricultural productivity for crop plants [2, 3]. Flowering time is controlled by several flowering-promoting pathways, including vernalization, autonomous, photoperiod, and gibberellin pathways [4]. Bolting is a key process in lettuce (*Lactuca sativa* L.) growth and development in which the lettuce plant sends up a stalk and produces seeds. The optimum lettuce growth temperature is between 15 and 20 ˚C; temperatures above 30 ˚C have resulted in early lettuce bolting [5], and early bolting decreases both quality and production. In our study, we used isobaric tags for relative and absolute quantitation (iTRAQ) to distinguish differentially expressed proteins [6, 7].

The study of phosphorylation regulation is important for elucidating the functional biology of plant proteins, and reversible protein phosphorylation plays critical roles in transducing stress signals according to coordinated intracellular responses. However, the molecular mechanism underlying lettuce bolting at high temperatures has not been studied. Protein phosphorylation is an important posttranslational modification [8]; many biological processes are regulated by protein phosphorylation, including transcription, metabolism, translation, protein degradation, homeostasis, cellular signaling and communication [9]. In Arabidopsis, the phosphorylation of ethylene response factor 110 regulates bolting time, and further physiological and biochemical studies have revealed that serine 62 phosphorylation of ethylene response factor 110 is linked to bolting time [10].

Many stress-related proteins are phosphorylated under diverse conditions, suggesting an important role of protein phosphorylation in response to various stresses. Phosphorylation can activate enzymes involved in starch synthesis and increase the physical interactions among those enzymes [11]. In potato tubers, hexose-phosphate levels increase, whereas glycerate-3-phosphate (3PGA), phosphoenolpyruvate, and adenine diphosphoglucose (ADPGlc) levels decrease under elevated temperatures. When temperatures reach above 30 ˚C, the overall activities of sucrose synthase and ADPGlc pyrophosphorylase decline slightly, whereas the activities of soluble starch synthase and pyruvate kinase remain unchanged. Elevated temperatures lead to increased rates of respiration, and the resulting decline in 3PGA levels inhibits ADPGlc pyrophosphorylase and starch synthesis [12]. High temperatures decrease starch synthesis by inhibiting starch biosynthetic enzymes while increasing respiration or decreasing sucrose mobilization, resulting in declining levels of precursors [12]. Furthermore, in higher plants, sucrose metabolism invertases hydrolyze sucrose into glucose and fructose. These molecules are related to photosynthesis and are used as nutrients, energy sources, and signaling molecules for plant growth, yield formation, and stress responses [13].

This study aimed to provide a comprehensive characterization of the lettuce phosphoproteome that could help elucidate the phosphorylation events underlying the molecular mechanism of early bolting in lettuce. Our study will provide new insight for the molecular breeding of lettuce to improve species resistance to high temperature and obtain new varieties.

## Materials and methods

### Plant materials and treatment

Seeds of lettuce (*Lactuca sativa* L.) GB-30, which is an easy bolting variety, were sown in a sand/soil/peat (1:1:1 v/v) mixture and grown at the Beijing University Agriculture Experimental Station under standard greenhouse conditions (14 h of light; 300–1300 μmol/($m^2$ s); 20 ± 2 ˚C during the day; 13 ± 2 ˚C at night; 10 h of darkness; and 50%-70% relative humidity). When the plants developed the sixth true leaf, they were divided into two groups. The control group was treated under standard greenhouse conditions with a light duration of 14 h, a darkness duration of 10 h, a light intensity of 300–1300 μmol/($m^2$ s), a temperature of 20 ± 2 ˚C during the day, a temperature of 13 ± 2 ˚C at night, and a relative humidity of 50%-70%. The

other group was exposed to high-temperature conditions, with a light duration of 14 h, a darkness duration of 10 h, a light intensity of 300–1300 μmol/($m^2$ s), a temperature of 33 ± 2 ˚C during the day, a temperature of 25 ± 2 ˚C at night, and a relative humidity of 50%-70%. After high-temperature treatment, stems from the control and high-temperature treatment groups were collected on days 0, 4, 8, 12, 20, and 32 (S1 Fig), frozen in liquid nitrogen, and stored at -80 ˚C for further physiological analysis. On day 32 (obvious bolting stage, as determined by our previous study) [6], we collected stem samples from the control and treated plants, froze them in liquid nitrogen, and stored them at −80 ˚C for further phosphoproteomic analysis.

## Protein extraction

Samples were homogenized in a liquid nitrogen-chilled cryogenic mill and then precipitated in 45 ml 10% (w/v) trichloroacetic acid/acetone containing 65 mM dithiothreitol (DTT) at -20 ˚C overnight. The homogenates were centrifuged at 7000 × $g$ for 20 min, and the pellets were rinsed with 40 ml ice-cold acetone and then centrifuged at 7000 × $g$ for 15 min. The pellets were washed three times with acetone and air-dried. Afterwards, 800 μL of SDT lysis buffer (4% SDS, 100 mM Tris/HCl, and 1 mM DTT, pH 7.6) was added to the dried pellets, and the samples were heated twice in boiling water for 15 min The samples were ultrasonicated (10 rounds of 100-W sonication for 10 s with 10-s intervals) and then centrifuged (20 min, 13400 rpm). The supernatants were collected, and the protein concentrations were measured using the BCA method.

## Phosphoprotein identification and quantitation

The MS/MS spectra were searched against a database containing 85159 protein sequences of *Lactucasativa* (Lactuca.Unigene.pep.fasta, constructed by our laboratory, December, 2018) using the MASCOT engine (version 2.0, Matrix Science, London, UK) with the Proteome Discoverer 1.4 (Thermo Electron, San Jose, CA, USA) plugin. The following parameters were used for phosphoprotein identification: enzyme = trypsin; max missed cleavages = 2; fixed modifications: carbamidomethyl (C), iTRAQ4/8 plex (N-term), iTRAQ 4/8 plex (K); and variable modifications: oxidation (M), iTRAQ 4/8plex (Y), phospho (ST)/phosphor (Y), peptide mass tolerance = 20 ppm, and fragment mass tolerance = 0.1 Da. The false discovery rate (FDR) was ≤0.01 with a decoy database searching strategy to filter negative identification at the protein, peptide, and modification levels. For each phosphorylation site, the probability that the site was truly phosphorylated was estimated based on the Ascore algorithm, and only sites with a phosphosite probability above 0.75 were considered to be truly phosphorylated.

Quantification of peptides was performed based on the strength of the reporter ion. One sample t-test was used to compare the relative abundance of each peptide between the control and high-temperature treatment groups, and fold changes >1.2 or <0.84 and *p*-values < 0.05 were considered statistically significant.

## GO annotation and KEGG pathway analysis

The functional category of each identified phosphoprotein was annotated with Blast2GO software (version 3.3.5). The biological pathways related to the identified phosphoproteins were mapped using the online Kyoto Encyclopedia of Genes and Genomes (KEGG) database (http://www.genome.jp/kegg). Using Fisher's exact test, statistically overrepresented GO categories and KEGG pathways were identified. The Benjamini-Hochberg correction for multiple testing was further used to correct the derived p-values. All enriched GO terms were selected based on *p*< 0.05. For the relative phosphoprotein and peptide abundance, hierarchical

clustering analysis was carried out using Cluster 3.0 (http://bonsai.hgs.jp/~mdehoon/software/cluster/software.htm) and Java Tree view software (http://jtreeview.sourceforge.net). A Euclidean distance algorithm was used to measure similarity, and an average linkage clustering algorithm was selected for clustering.

## Motif analysis

The phosphorylation motif sets were extracted using a motif-X algorithm (http://motif-x.med.harvard.edu/motif-x.html). The background was the uploaded lettuce proteome, the motif width was 13, the occurrence was 20, and the significance was 1 $e$ -6. The motif was extracted separately on Ser, Thr, and Tyr sites at position 7.

## Measurement of sugar components and starch

The contents of sugar components and starch in lettuce were measured with proper modification [14, 15].

## Statistical analysis

The experiment was performed in triplicate. For physiology and proteome analyses, three different stems were pooled together as one biological sample, and this was done three times to produce three independent biological replicates. The SPSS 17.0 software package (SPSS, Chicago, USA) was used for statistical analysis of differences between treatment groups. The data were analyzed by Duncan's test to compare the differences between the experimental groups at $^*p \leq 0.05$ and $^{**}p \leq 0.01$. The data are presented as the mean ± SD.

# Results

## Phosphoproteomic profiles of control and high temperature-treated lettuce

Our previous study identified several protein kinases responsive to high temperature, suggesting that phosphorylation events play an important role in the responses of plants to high temperature, and the stems of lettuce in the high-temperature treatment group were significantly longer than those of lettuce in the control group [6]. To understand the molecular basis of bolting, we conducted phosphoproteomic analysis of lettuce stems during the bolting period induced by a high temperature (33 ˚C) and a control temperature (20 ˚C). A total of 3,814 phosphorylation sites in 1,766 phosphopeptides from 987 phosphoproteins were identified (S1 Table, Fig 1a). The highest percentage of these phosphopeptides contained two sites (45.4%), followed by single sites (27.6%) and three or more sites (27%) (Fig 1b). Regarding the pattern of identified phosphorylation sites, the highest percentage of observed phosphosites were Ser residues (84.6%), followed by Thr residues (13.8%) and Tyr residues (1.6%), similar to the eukaryotic phosphorylation classes (Fig 1c).

## Biological categories of significantly changed phosphopeptides after high-temperature treatment

To further investigate the significant change in phosphopeptide biological function after high-temperature treatment, one sample t-test was used to analyze peptide abundance, and biological process (BP), molecular function (MF), cellular component (CC), and KEGG pathways of the significantly changed peptides were investigated. Of the 1766 phosphopeptides, 217 phosphopeptides (116 upregulated and 101 downregulated) showed a change in abundance at p< 0.05 and a fold change > 1.2 or <0.84 after high-temperature treatment (S2 Table, Fig 2a). For BP analysis, 14 categories were annotated at level two, and the majority of BP classes were

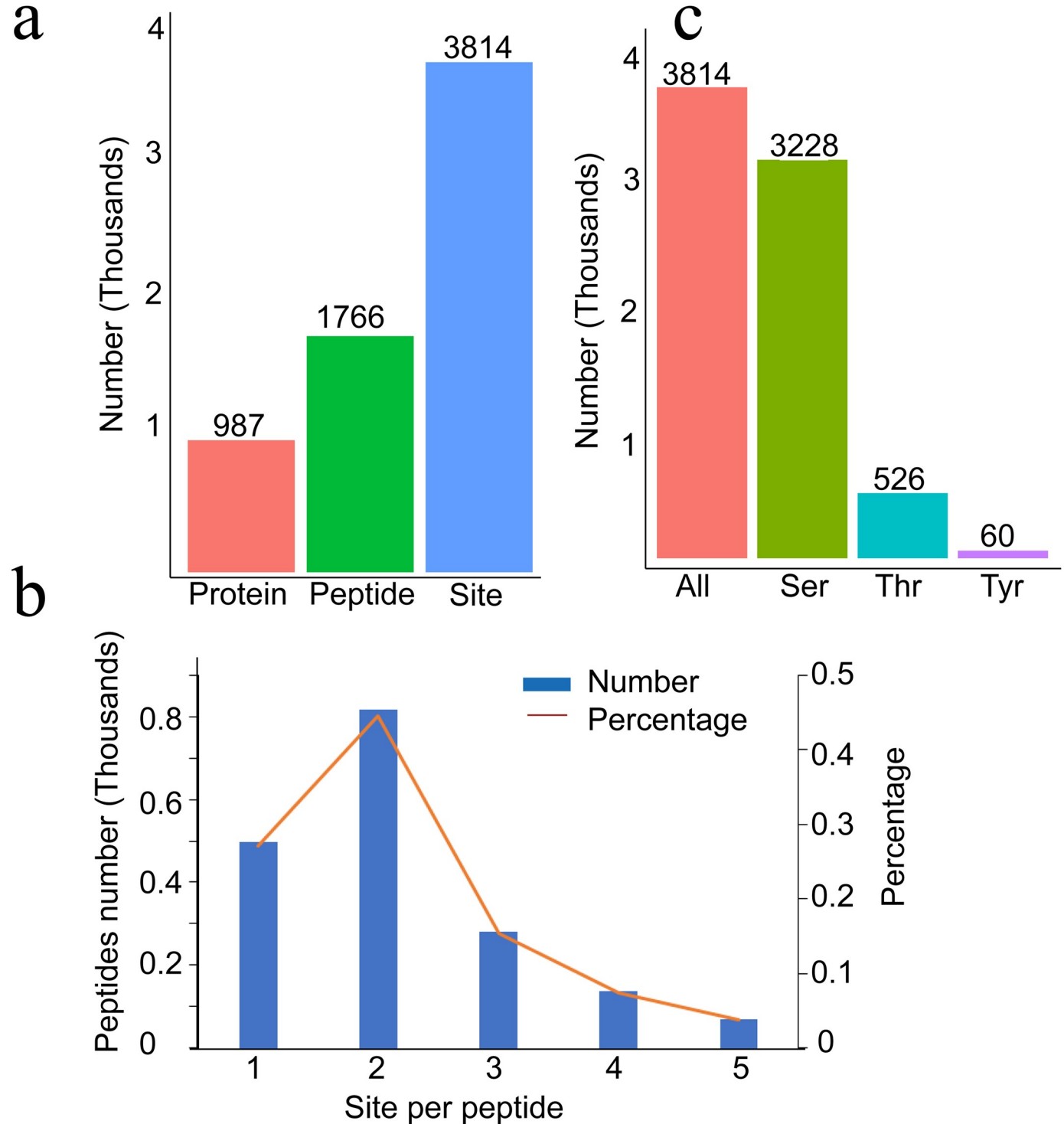

**Fig 1. Phosphorylation statistics.** (a) Identified phosphoproteins, (b) phosphopeptide site analysis, and (c) patterns of identified phosphorylation sites.

associated with cellular process, metabolic process, biological regulation process, localization and response to stimulus. MF classes were involved in catalytic and transporter activity, and CC classes were mostly involved in membrane, cell, cell part, membrane part, and organelle (Fig 2b).

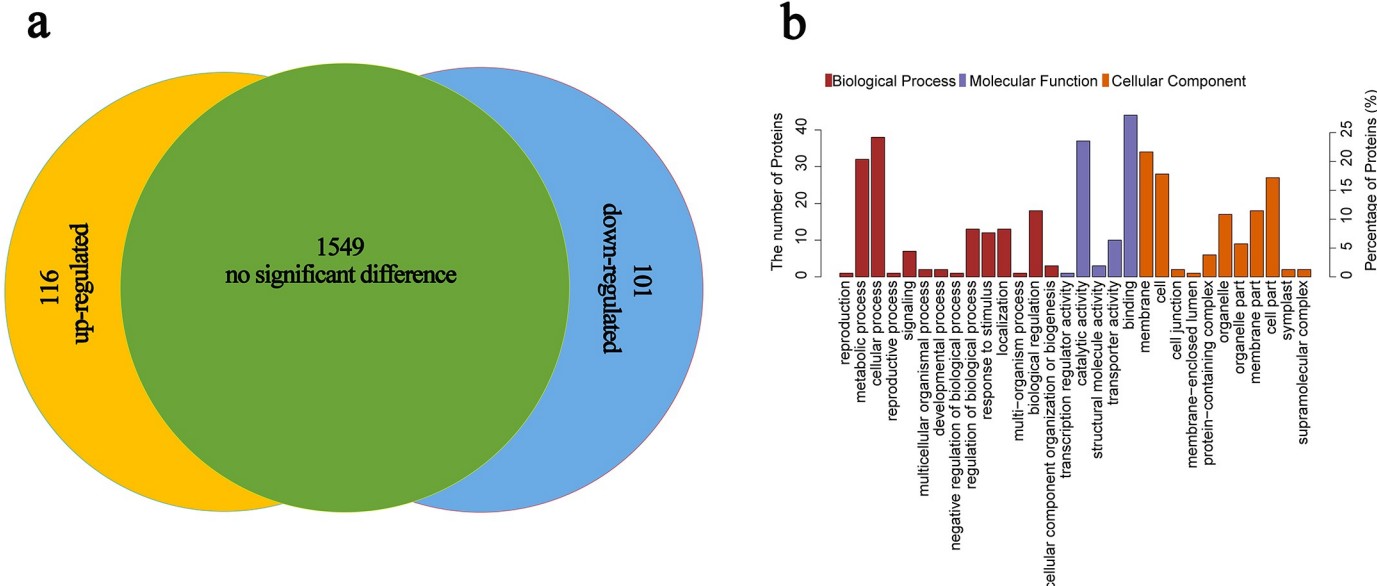

**Fig 2. Distribution and ClueGO analysis of differential phosphopeptides in lettuce under high-temperature treatment. (a)** Distribution analysis of differential phosphopeptides and (**b**) GO annotation for molecular function, biological process, and cellular component pathways.

Using Fisher's exact test, we found that the majority of peptides that showed changes in abundance after high-temperature treatment were strongly associated with BP pathways of transport, localization, homeostasis, signal transduction by protein phosphorylation, and especially water homeostasis. Some of these peptides, such as dehydrin Rab18-like (4.27 times), aquaporin PIP2-7 (1.62 times), and aquaporin PIP2-1-like phosphoproteins (1.38 times), exhibited large increases in expression levels. The most enriched MF categories were catalytic, transport, phosphotransferase, and protein kinase activity, and the most enriched CC categories were membrane (Fig 3).

In total, 104 signaling pathways with significantly changed peptides were mapped, of which spliceosome, starch and sucrose metabolism, RNA transport, protein processing in endoplasmic reticulum, phosphatidylinositol signaling system, phospholipase D signaling pathway, apoptosis, protein export, tight junction, and glycerolipid metabolism were the top ten identified signaling pathways (Fig 4a). Fisher's exact test for enrichment revealed that high temperature influenced phosphoprotein activity associated with metabolism, such as starch and sucrose metabolism, glycerolipid metabolism, and methane metabolism. The expression levels of phosphoprotein probable alpha, alpha-trehalose-phosphate synthase [UDP-forming] 7 (1.35-fold), glucose-6-phosphate isomerase 1, chloroplastic (1.32-fold), probable sucrose-phosphate synthase 1 (0.63-fold), beta-amylase 1, chloroplastic (0.75-fold), which is implicated in starch and sucrose metabolism, and diacylglycerol kinase 4-like (1.29-fold), which is involved in glycerolipid metabolism, were changed. Protein export, endocrine, and other factor-regulated calcium reabsorption were significantly affected (S2 Table, Fig 4b). The abundance of five proteins associated with biosynthesis in starch and sucrose metabolism, namely, trehalose 6-phosphate synthase/phosphatase (TPS), glucose-6-phosphate isomerase (GPI), two sucrose-phosphate synthases [EC:2.4.1.14], and beta-amylase [EC:3.2.1.2], was found to be increased, of which four proteins with differential abundance were involved in amino sugar and nucleotide sugar metabolism signal transduction pathways (S2 Table).

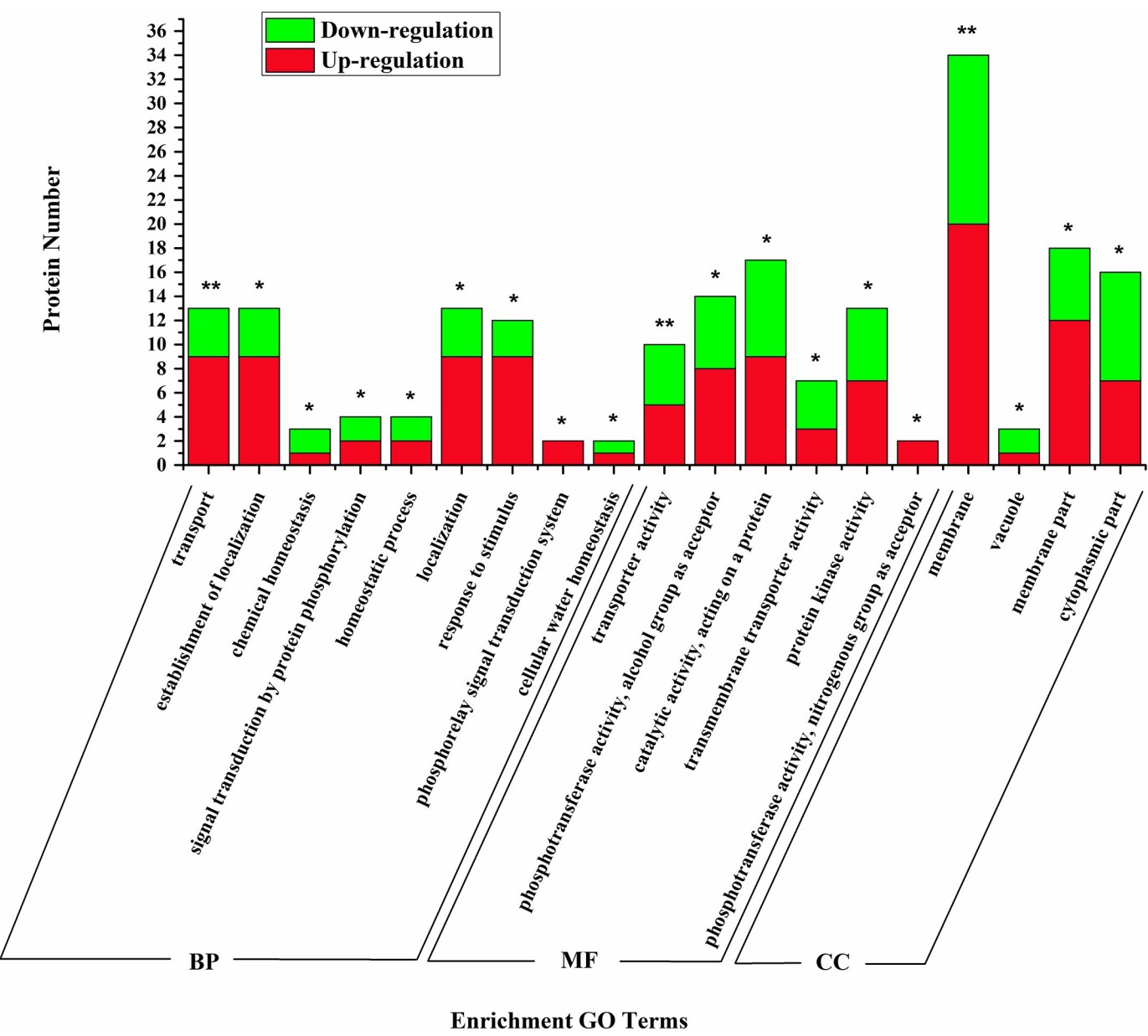

**Fig 3. GO enrichment analysis of phosphopeptides with differential abundance.**

## Upstream enzymes and their involvement in biological pathways based on the motif-X algorithm

Motif analysis is used to identify specific phosphorylation kinases based on the position of the specific phosphorylation modification on the phosphorylated peptide; that is, certain kinases phosphorylate specific forms of protein substrates at specific sites. Based on this, we used the online software Motif-X (http://motif-x.med.harvard.edu/motif-x.html) to predict the motif specificities of these phosphoproteins based on the identified phosphorylation sites [17]. Using the Motif-X algorithm, we found 16 kinds of motifs, and these extracted motifs were classified into four kinase classes: proline-directed, acidic, basic, and others and tyrosine (Fig 5a). The

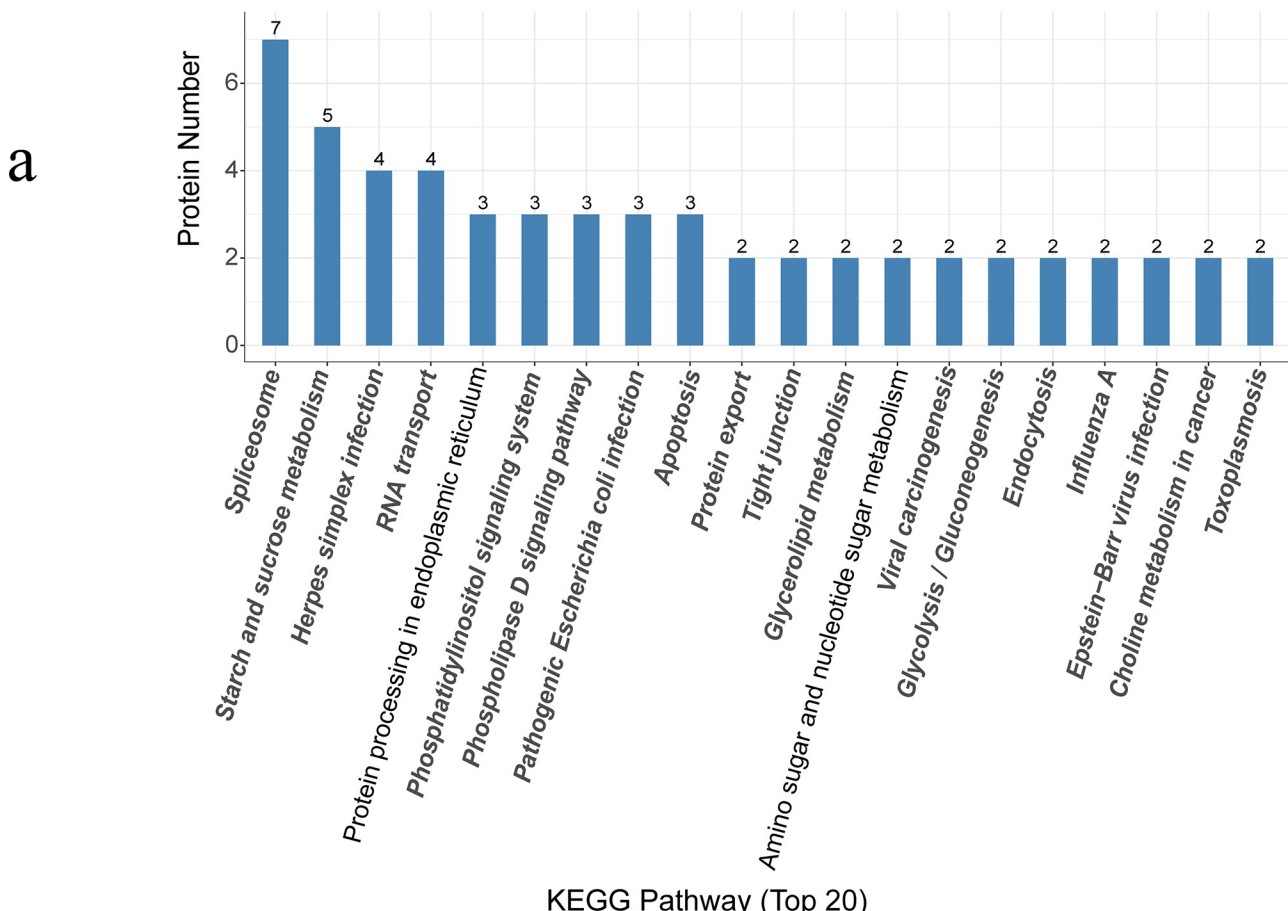

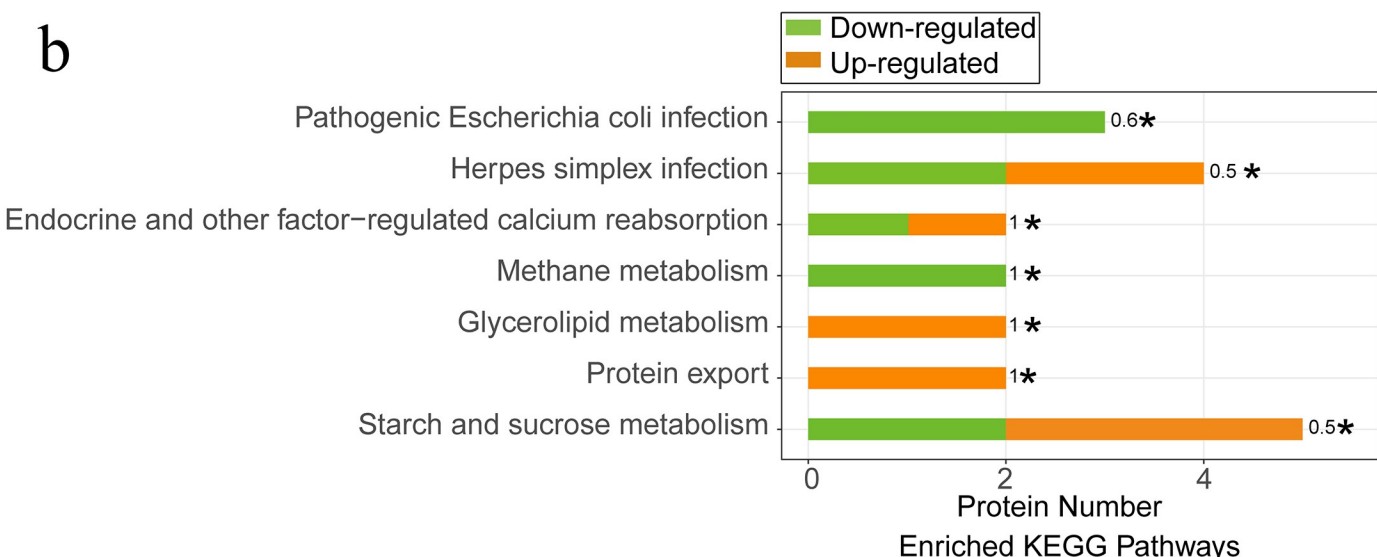

**Fig 4. KEGG annotation and pathway enrichment.** (**a**) The top 20 identified signaling pathways associated with peptides showing significant changes in abundance. (**b**) Phosphoprotein activity associated with metabolism under high-temperature treatment determined by Fisher's exact test.

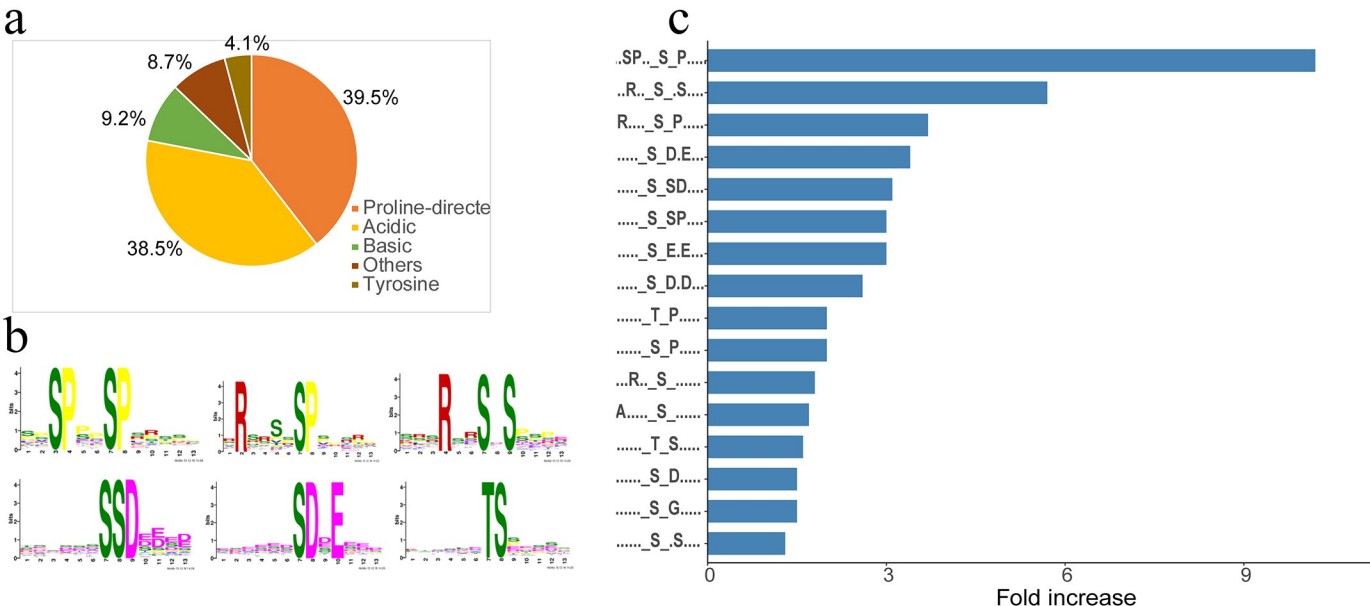

**Fig 5. Phosphorylation-specific motifs identified by the Motif-X algorithm.** (**a**) The four kinase classes based on the Motif-X algorithm. (**b**) Motif analysis of phosphorylation sites. (**c**) Several kinases may play key roles in lettuce bolting.

proline-directed motif (..SP..[s]P, [s]P, [t]P) was the dominant class, followed by acidic ([s]D. E, [s]SD, [s]E.E, [s]DD), basic (R..[s].[s], R..[s]), and others and tyrosine (T[S]) (Fig 5b). A certain motif is usually recognized by specific kinases; for example, proline-directed motifs are associated with MAK kinase, and several acidic classes are casein kinase II motifs. These kinases may play key roles in lettuce bolting (Fig 5c).

## Sugar components and starch assays

To verify the effects of late bolting, we measured the contents of sugar components and starch after high-temperature treatment for 0–32 days (Fig 6) and found that the contents of galactose and sucrose were significantly lower in the high-temperature lettuce than in the control lettuce from day 8 to day 32 (Fig 6a and 6d); glucose, fructose and starch levels were significantly higher in the high-temperature treatment than in the control from day 20 to day 32 (Fig 6b, 6c and 6e).

## Discussion

Protein phosphorylation plays an important role in multiple cellular functions involved in temperature responses [16, 17]. Phosphoproteomic studies on the molecular basis of regulatory mechanisms in plants will improve our understanding of fundamental, complex biological processes and provide information with potential agricultural applications. This study suggests that high temperature effects cell growth, differentiation, and apoptosis via metabolic processes, especially starch and sucrose metabolism and that this effect is regulated by protein phosphorylation. We hypothesize that along sugar components and starch, saccharides play crucial roles in high temperature-induced bolting in lettuce.

Protein phosphorylation requires upstream enzymes to recognize substrate-specific motifs; therefore, the recognition of modified motifs can enable us to identify these upstream enzymes and elucidate the biological pathways in which they are involved. We identified 16 kinds of

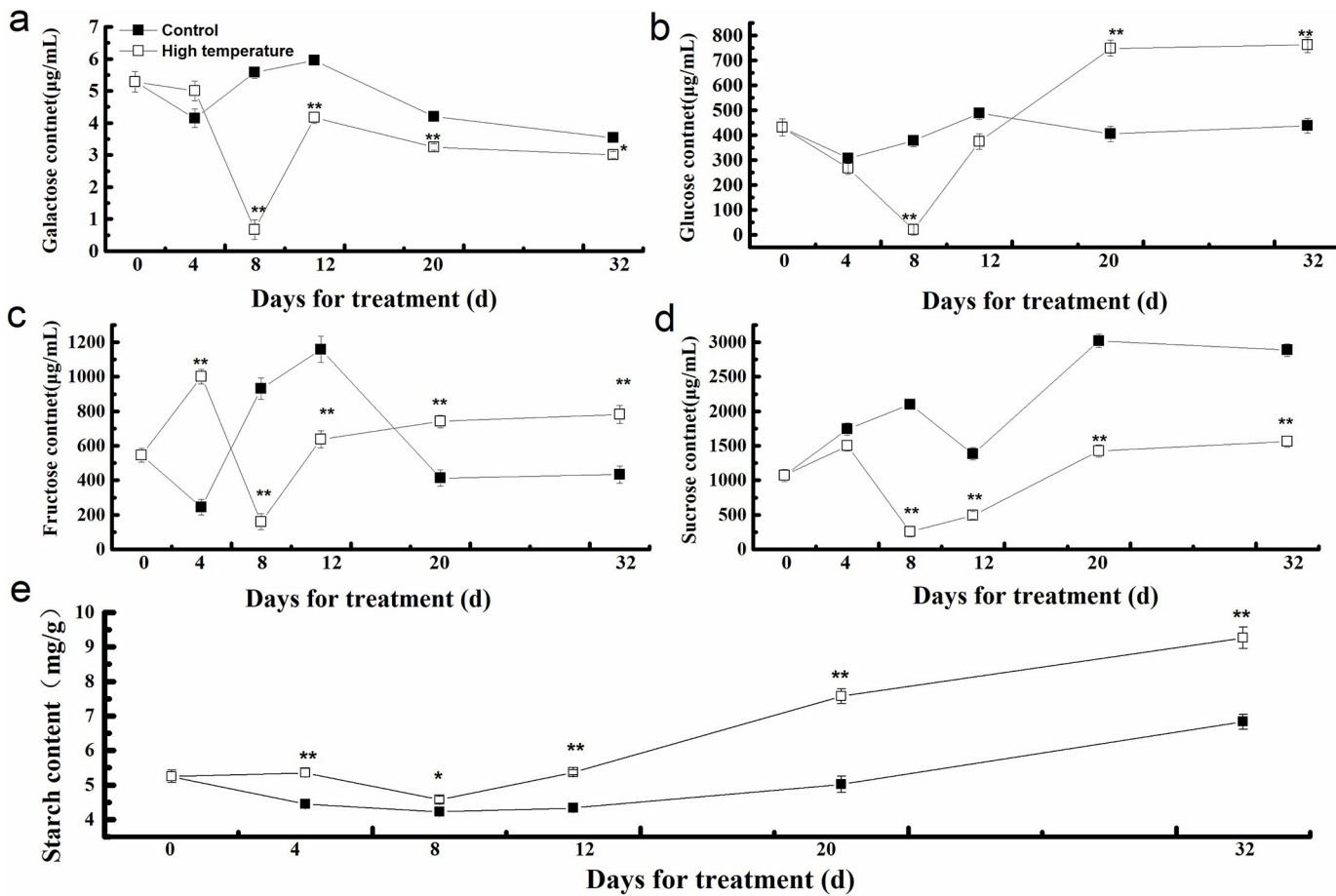

**Fig 6. Changes in the contents of sugar components and starch in lettuce during bolting induced by high temperature.** (**a**) Galactose content. (**b**) Glucose content. (**c**) Fructose content. (**d**) Sucrose content. (**e**) Starch content.

extracted motifs that could be classified into four kinase groups: proline-directed, acidic, basic, and others and tyrosine. A certain motif is usually recognized by specific kinases' proline-directed motifs are recognized by MAK kinases, and several acidic motifs are recognized by casein kinase II. These kinases play key roles in cell proliferation and lettuce growth.

Abiotic stresses, such as drought, salinity, or temperature variation, impair the productivity of all major crops and reduce average yields [18]. Temperature is an essential abiotic factor for plant growth, metabolism and productivity. Owing to the large water amounts in plant tissues, cell water behavior under different temperatures is key for cell survival. High temperatures provoke heat stress and accelerate respiration and photosynthetic rates [18, 19], leading to reactive oxygen species overproduction and affecting plant productivity. High temperatures increase plasma membrane fluidization, and these changes affect the activity of membrane-associated proteins, such as ATPases, ion transporters, H+ pumps, and protein channels. Consistent with this, the protein expression of aquaporin PIP2-1-like and dehydrin Rab18-like, which are associated with osmoregulation, was upregulated by high-temperature treatment. Fluidization changes are primary stimuli that trigger the flux of second messengers (e.g., $Ca^{+2}$) to activate specific heat responses. These stimuli include calcium-dependent protein kinases and MAP kinases. Protein kinases are necessary for protein phosphorylation. In eukaryotes, protein phosphorylation sites are predominantly serine, threonine, and tyrosine residues, and

they are phosphorylated by specific protein kinases. MAPK is a prevalent Ser/Thr protein kinase that is highly conserved in all eukaryotes. MAPK pathways have significant functions in plants, such as the immune response, developmental processes, abiotic stress responses, and hormonal regulation [20].

It has been reported that in alfalfa cells, MAP kinase is the first molecule to respond to heat [21]. However, relatively little research on the response of MAP kinase signaling to heat in plants has been conducted. In our study, MAPK kinase showed a trend of upregulated expression at high temperatures. Another study showed that MAPK is involved in the metabolic process of abscisic acid (ABA), a plant growth-inhibiting hormone. Thus, ABA can also control MAPK generation [22]. MPK12 is activated by ABA [23], and Ichimura and coworkers reported that under abiotic stress conditions, including cold temperatures, low humidity, and osmotic shock, ABA can activate Arabidopsis MPK4 and MPK6 [24]. Therefore, we conclude that high temperatures caused MAPK expression upregulation, which resulted in an increase in ABA levels. Therefore, lettuce was prematurely transitioned to the reproductive growth period.

Casein kinase (CK) 1 and 2 are vital kinases for plant growth. However, the two kinases belong to different kinase families and act on different hormones. CK1 plays major regulatory roles in many cellular processes, including DNA processing and repair, proliferation, cytoskeleton dynamics, vesicular trafficking, apoptosis, and cell differentiation [25]. Some have suggested that CK1 can affect ethylene synthesis [26]; researchers have found that the absence of CK1.8 (a CK1 isoform) causes ACS5 (a key enzyme in ethylene synthesis) to accumulate in large quantities, leading to ethylene overexpression. Based on this conclusion, the downregulation of CK1 expression at high temperature leads to ethylene overexpression and early bolting/flowering. CK2 is a pleiotropic and highly conserved serine or threonine protein kinase that plays a key role in cell division and cell expansion, as confirmed in Arabidopsis [27]. Furthermore, CK2 has been shown to regulate developmental and stress response pathways in functional studies on various plant species [28]. Moreover, CK2 can modulate auxin-responsive gene expression by modulating the stabilization of AXR3, which is a member of the AUX/IAA family of auxin transcriptional repressors [29]. We hypothesize that CK2 expression upregulation disrupted the stabilization of AXR3 to degrade the auxin inhibitory factor and that lettuce stem growth occurred earlier.

Sugar plays a profound physiological role in plant growth and development. In chickpea plants, heat stress results in a decrease in the activities of sucrose-phosphate synthase and sucrose synthase, causing a shortening of the normal growth cycle and a reduction in fruit yield [30]. Consistent with this, the expression of sucrose non-fermenting 1 (SNF1)-related protein kinase (SnRK), which is a critical kinase for the restoration of homeostasis in plants and promotion of plant stress tolerance during environmental stress and carbohydrate deficiency [31], was significantly upregulated after high-temperature treatment. Usually, SnRK inhibits vegetative growth and facilitates the transition of plants to the generative stage [32]. In this study, the upregulation of SnRK expression contributed to a reduction in the level of the soluble sugars galactose and sucrose in the leaves of the high-temperature treatment group, probably because the soluble sugars were mainly transported from the leaves to the stem to provide energy for bolting. Furthermore, the expression of proteins with carbohydrate synthesis activity was upregulated after high-temperature treatment, such as alpha-trehalose-phosphate synthase and glucose-6-phosphate isomerase 1, chloroplastic, leading to increases in the contents of glucose, fructose, and starch in the leaves of high-temperature-treated lettuce. High levels of sugar lead to the activation of trehalose-6-phosphate and TOR protein kinase, and the upregulation of trehalose-6-phosphate and TOR protein kinase expression is important for cell growth in plants. It has been reported that trehalose-6-phosphate plays an essential

role in the growth of A. *thaliana* [33, 34], and a lack of trehalose-6-phosphate or TOR kinases inhibits growth and transition to the reproductive phase [35, 36]. Additionally, sugar has been shown to reduce water movement, which affects leaf growth [37]. Here, we found that the expression of aquaporin PIP2-1-like and dehydrin Rab18-like, which are associated with water transport, was also upregulated in high-temperature-treated lettuce.

In summary, high temperatures mainly affect carbohydrate synthesis enzyme activity, leading to variations in sugar concentrations. Sugars not only provide storage in plants and provide energy for high temperature-induced bolting but also act as signaling molecules. Sugars further activate MAK kinase, casein kinase, trehalose-6-phosphate and TOR protein kinase, which play key roles in cell division and growth, and ultimately accelerate the rate of transition from vegetative growth the reproductive growth stage in lettuce, contributing to bolting.

In this study, we identified several proteins associated with starch and sucrose metabolism under high temperatures, and the expression of most proteins associated with starch and sucrose metabolism was significantly upregulated. Carbohydrates are sources of energy for plant growth and development and are raw materials for certain tissue ingredients.

## Conclusions

Quantitative phosphoproteomic analysis led to the identification of key phosphopeptides and phosphoproteins involved in the response of lettuce to high temperature. Most phosphoproteins that showed changes in abundance were associated with the metabolic process; correspondingly, the main related molecular functions were catalytic activity and transporter activity. High temperature influenced phosphoprotein activity associated with metabolism, especially starch and sucrose, glycerolipid, and methane metabolism. Our results suggested that the majority of peptides that showed changes in abundance after high-temperature treatment were significantly associated with BP pathways of transport, localization, homeostasis, signal transduction by protein phosphorylation, and especially water homeostasis. Proline-directed, acidic, basic, and others and tyrosine kinases were predicted to be involved in signal transduction and metabolism regulation. The proline-directed motif class including..SP..[s]P, [s]P, [t]P was the dominant class, and proline-directed motifs associated with MAK kinase and several acidic motifs related to casein kinase II motif were necessary for lettuce bolting.

Our study showed that the contents of soluble sugars, including galactose, sucrose, glucose, and fructose, were changed during bolting under high temperature in lettuce leaves; the starch content in the leaves was significantly increased during late bolting. We concluded that the lettuce had to shorten its growth cycle and reduce vegetative growth owing to changes in the contents of starch and soluble sugar, which led to early bolting/flowering. Future studies should focus on identifying the kinases that phosphorylate these phosphoproteins associated with starch and sucrose metabolism in response to high temperature.

## Supporting information

**S1 Table. Qualitative analysis of phosphorylated peptides.**
(XLSX)

**S2 Table. Quantitative analysis of significantly differentiated phosphorylated peptides under high-temperature treatment.**
(XLSX)

**S1 Fig.**
(TIF)

## Acknowledgments

The mass spectrometry proteomics data have been deposited to the ProteomeXchange Consortium (http://proteomecentral.proteomexchange.org) via the iProX partner repository with the dataset identifier PXD014208. We are grateful to Rui Yan and Jie Kang from Shanghai Applied Protein Technology for proteomic and bioinformatics analysis. We are also grateful to Jianke Li from Chinese Academy of Agricultural Science for advice on preparing paper.

## Author Contributions

**Conceptualization:** Shuangxi Fan.

**Data curation:** Lifeng Meng, Jinghong Hao.

**Formal analysis:** Shaowei Lu.

**Investigation:** Xiaoxiao Qin, Panpan Li.

**Methodology:** Shaowei Lu, Yanchuan Sun.

**Resources:** Jinghong Hao, Shuangxi Fan.

**Software:** Lifeng Meng.

**Validation:** Yanchuan Sun.

**Writing – original draft:** Xiaoxiao Qin.

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
