## [Decision Letter · Decision Letter 0]

28 Aug 2020

PONE-D-20-23881

Phosphoproteomic Analysis of Lettuce (Lactuca sativa L.) Reveals Starch and Sucrose Metabolism Functions during Bolting Induced by High Temperature

PLOS ONE

Dear Dr. qin,

Thank you for submitting your manuscript to PLOS ONE. After careful consideration, we feel that it has merit but does not fully meet PLOS ONE’s publication criteria as it currently stands. Therefore, we invite you to submit a revised version of the manuscript that addresses the points raised during the review process.

We look forward to receiving your revised manuscript.

Kind regards,

Yuan Huang

Academic Editor

PLOS ONE

2. Please amend the manuscript submission data (via Edit Submission) to include author Jinghong Hao

Reviewers' comments:

Reviewer's Responses to Questions

**Comments to the Author**

1. Is the manuscript technically sound, and do the data support the conclusions?

Reviewer #1: Partly

Reviewer #2: Yes

2. Has the statistical analysis been performed appropriately and rigorously? 

Reviewer #1: I Don't Know

Reviewer #2: N/A

3. Have the authors made all data underlying the findings in their manuscript fully available?

Reviewer #1: No

Reviewer #2: Yes

4. Is the manuscript presented in an intelligible fashion and written in standard English?

Reviewer #1: No

Reviewer #2: No

5. Review Comments to the Author

Reviewer #1: The Manuscript entitle as " Phosphoproteomic Analysis of Lettuce (Lactuca sativa L.) Reveals Starch and Sucrose Metabolism Functions during Bolting Induced by High Temperature " by Qin Tao et al, described the use of Phosphoproteome to compare the change in different temperature during the bolting time in lettuces. The research seems to be novel and interesting. However, fragmentary results were not well-organized to lead the mechanism of starch and sucrose metabolism.

There is room for improvement. Below are summarized major and minor of the manuscript that should be improved.

Major:

1. The results in this manuscript are too simple. For example, in the paragraph “Phosphoproteome profile of control and high-temperature treated lettuce”, the author did not describe any tissues to compare, and all the details can not find in the Materials and Methods. It is hard to understand the results of this manuscript.

2. I can not understand what relationship of this paragraph “Upstream enzymes and their biological pathway involvement 194 based on the motif-X algorithm” with the first two paragraphs.

3. In the paragraph “Sugar components and starch assays”, the author measured the content of different kinds of sugar and starch. The results showed that the kinds of sugar are different in the different treatments. However, it was not the direct evidence to show their relationship with blotting in lettuce.

Minor:

P7 L138 3814 > 3,814 1766 > 1,766

P7 L140 double > two

P7 L150 quantify > analyze

P7 L153 chaned > change on

P9 L209 between > during

P9 L211 lettuce than the control lettuce between the 20-32days > treatment than the control during the 20-32days

The manuscript should be well reorganized, and all results should be logical and realible.

Reviewer #2: The authors describe a global effort, by iTRAQ-based phosphoproteom and the content of Starch and Sucrose, in order to explore a molecular mechanism that high temperature promotes premature bolting. 217 significantly different phosphoproteins

were indentified. Most identified phosphoproteins were involved in starch and

sucrose metabolism metabolic process; and the content of related sugar and starch in different treatment in lettuce were support the conclusion. The study has been properly carried out and results display an adequate level of novelty and interest in the field.

1) In this study, the sugar and starch play important role during lettuce bolting. The authors had little discussion about it(line273-277). However more deep discussion is need in this article, such as what’s the detail function that the sugar and starch play in other species. Or the sugar and starch include many kinds of chemical compound, what is the function of them in plant development?

2. As the authors describe in this article (lin124-126), for physiology and proteome analyses, three different stems were pooled together as one biological sample, and this was done three times to produce three independent biological replicates. There plants for physiology and proteome as one biological was not enough. Maybe it was not reflect the real situation, especially for physiology. So can you cite other related research for more support.

3. Most abundance changed phosphoproteins were associated with the metabolic process; correspondingly, the main molecular functions were catalytic activity and transporter activity. The most important signaling pathways was starch and sucrose metabolism, and high

temperature influenced phosphoprotein activity especially associated with starch and sucrose

13 metabolism. The authors should dig the logical relationship between The molecular function about catalytic activity and transporter activity, and starch and sucrose metabolism in discussion. Such as which enzymes with catalytic activity and transporter activity play the role in metabolism of starch and sucrose.

4. The authors had better add the photo about On day 0, 4, 8, 12, 20, 32 after high temperature treatment and control.

5. The English of your manuscripts must be improved before resubmission. I strongly suggest that you get the assistant from a native English speaker.

6. PLOS authors have the option to publish the peer review history of their article (what does this mean?). If published, this will include your full peer review and any attached files.

Reviewer #1: No

Reviewer #2: No

---

## [Author Response · Author response to Decision Letter 0]

5 Nov 2020

Response to the Reviewers’ comments:

Both Reviewers were very supportive of our manuscript. We thank them for their time and efforts to review our manuscript.

Reviewer #1: The Manuscript entitle as " Phosphoproteomic Analysis of Lettuce (Lactuca sativa L.) Reveals Starch and Sucrose Metabolism Functions during Bolting Induced by High Temperature " by Qin Tao et al, described the use of Phosphoproteome to compare the change in different temperature during the bolting time in lettuces. The research seems to be novel and interesting. However, fragmentary results were not well-organized to lead the mechanism of starch and sucrose metabolism.

There is room for improvement. Below are summarized major and minor of the manuscript that should be improved.

Response: Thanks the reviewer for the valuable advice. We have extensively revised our manuscript, improved the writing style and restructured the Methods sections, Results and Discussion, also we have revised the duplicate copyright statement in the manuscript.

Major:

1. The results in this manuscript are too simple. For example, in the paragraph “Phosphoproteome profile of control and high-temperature treated lettuce”, the author did not describe any tissues to compare, and all the details can not find in the Materials and Methods. It is hard to understand the results of this manuscript.

Response: Thank you for pointing this out. We have added more details in the Materials and Methods section and added the description with the following statement: “The control group was treated under the standard greenhouse conditions, the light time is 14 h and dark time is 10 h, light intensity is 300-1300 μmol/(m2 s), the temperature is 20 ± 2 °C during the day; 13 ± 2 °C at night, the relative humidity is 50%-70%. The other group was treated with high temperatures conditions, the light time is 14 h and dark time is 10 h, light intensity is 300-1300 μmol/(m2 s), the temperature is 33 ± 2 °C during the day; 25 ± 2 °C at night, the relative humidity is 50%-70%. After high temperature treatment, stems from control and high-temperature treatment were collected on 0 day, 4 day, 8 day, 12 day, 20 day, 32 day”.

2. I can not understand what relationship of this paragraph “Upstream enzymes and their biological pathway involvement 194 based on the motif-X algorithm” with the first two paragraphs.

Response: Thank you for pointing this out. We altered our previous sentence and explained the relationship of upstream enzymes and their biological pathway involvement based on the motif-X algorithm with the following statement: Motif analysis is to determine the specific phosphorylation kinase based on the position of the specific phosphorylation modification on the phosphorylated peptide, that is, certain kinases will undergo specific phosphorylation modification for specific forms of protein substrates. Based on this, we used online software Motif-X (http://motif-x.med.harvard.edu/motif-x.html) to predict the motif specificity of these phosphoproteins based on the identified phosphorylation sites [1].

[1] Hu X, Wu L, Zhao F, Zhang D, Li N, Zhu G, Li C, Wang W. Phosphoproteomic analysis of the response of maize leaves to drought, heat and their combination stress. Front. Plant Sci. 2015;6: 298

3. In the paragraph “Sugar components and starch assays”, the author measured the content of different kinds of sugar and starch. The results showed that the kinds of sugar are different in the different treatments. However, it was not the direct evidence to show their relationship with blotting in lettuce. 

Response: We agreed with the comments of reviewer. The change of sugar content is not the direct cause of bolting, but the content of sugar and starch is an important indicator when plants response to heat stress, there are many works demonstrate that the interconversion of sugar and starch serves as protectant against abiotic stress[1]. In addittion, sugar and starch can act as signaling molecules to regulate multitude process such as growth, development, aging and other plant organs troughout the life cycle of the plant. Recent studies have indicated that carbohydrate level mainly affect photosynthesis in leaves, plays a key role in lateral shoot development [2]. Sugars can also affect cell division and cell cycle stages, causing the abnormal growth of vegetative, and reproductive organs of plants occurred [3]. Blotting in lettuce is a process that actually shortened vegetative growth and enters reproductive growth in advance. This is a complex developmental process and regulated by many factors. Here, using a iTRAQ phosphoproteomic, we found hihg temperature stess mainly influence starch and sucrose metabolism process, and many associated kinase were up-regulated after high temperature stress, for example, MAP kinases, SNF1-related protein kinase and TOR protein kinase, these kinase play important roles in regulating the vegetative growth transfer to reproductive growth in plant. Furthermore, some key enzymes implicated in sugars synthes, such as trehalose-phosphate synthase, glucose-6-phosphate, were up-regulated after hihg temeture stress. Hence, together with our sugar components and starch assays, we hypothesis that high temperature influence blotting in lettuce via modification of starch and sucrose metabolism.

[1] Dong S , Beckles D M . Dynamic changes in the starch-sugar interconversion within plant source and sink tissues promote a better abiotic stress response. Journal of Plant Physiology, 2019.

[2] Astrid W. Transitioning to the Next Phase: The Role of Sugar Signaling throughout the Plant Life Cycle. Plant Physiology, 2018, 176(2):1075.

[3]Wang L, Ruan YL. Regulation of cell division and expansion by sugar and auxin signaling. Front Plant Sci. 2013;4:163. https://doi.org/10.3389/fpls.2013.00163

Minor:

P7 L138 3814 > 3,814 1766 > 1,766

P7 L140 double > two

P7 L150 quantify > analyze

P7 L153 chaned > change on

P9 L209 between > during

P9 L211 lettuce than the control lettuce between the 20-32days > treatment than the control during the 20-32days

The manuscript should be well reorganized, and all results should be logical and realible.

Response: Thank you very much for your careful review and pointing this out. We corrected our mistakes.

Reviewer #2: The authors describe a global effort, by iTRAQ-based phosphoproteom and the content of Starch and Sucrose, in order to explore a molecular mechanism that high temperature promotes premature bolting. 217 significantly different phosphoproteins were indentified. Most identified phosphoproteins were involved in starch and sucrose metabolism metabolic process; and the content of related sugar and starch in different treatment in lettuce were support the conclusion. The study has been properly carried out and results display an adequate level of novelty and interest in the field.

Response: Thanks very much for the reviewers affirming of our work and providing us with helpful advices for the paper.

1) In this study, the sugar and starch play important role during lettuce bolting. The authors had little discussion about it (line273-277). However more deep discussion is need in this article, such as what’s the detail function that the sugar and starch play in other species. Or the sugar and starch include many kinds of chemical compound, what is the function of them in plant development?

Response：It has been reported that the plant during the period of receiving biological or abiotic stress, in response to stress, free sugars will reduce, and sugar as an energy source and the structure composition, is essential for plant growth and metabolism(F. Rook et al, Sugar and ABA response pathways and the control of gene expression).The results of Bavita et al. Showed that wheat ripened earlier after high temperature treatment, and the content of free sugars and starch decreased significantly in response to high temperature stress(Bavita Asthir, Genotypic Variation for High Temperature Tolerance in Relation to Carbon Partitioning and Grain Sink Activity in Wheat). Furthermore, sugar acts as signaling molecules to regulate development of plant growth. We have revised the relevant part of the discussion.

2. As the authors describe in this article (lin124-126), for physiology and proteome analyses, three different stems were pooled together as one biological sample, and this was done three times to produce three independent biological replicates. There plants for physiology and proteome as one biological was not enough. Maybe it was not reflect the real situation, especially for physiology. So can you cite other related research for more support.

Response：In this study, we randomly sampled two sets of samples grown under high temperature treatment and control. In order to avoid biological errors, we randomly selected more than three plants in each period, chopped up the samples from the same period, and mixed them randomly. During the test, each sample was tested for three times, avoiding the data gap caused by the test error. The article “Metabolomic Proﬁling of Soybeans (Glycine max L.) Reveals the Importance of Sugar and Nitrogen Metabolism under Drought and Heat Stress” used the same sampling method. “A separate growth chamber contained pots that were exposed to high temperatures (43/35◦Cwith50%humidity) with water saturated soils; these were the third ‘heat’ group. After the seventh day of treatments, young trifoliate leaves from developmentally matched plants (number of plants >3) were harvested at noon in bulk from one pot as one replication unit, based on previously published studies in Soybean, and were immediately snap-frozen in liquid nitrogen and stored at−80◦C.”

3. Most abundance changed phosphoproteins were associated with the metabolic process; correspondingly, the main molecular functions were catalytic activity and transporter activity. The most important signaling pathways was starch and sucrose metabolism, and high temperature influenced phosphoprotein activity especially associated with starch and sucrose metabolism. The authors should dig the logical relationship between the molecular function about catalytic activity and transporter activity, and starch and sucrose metabolism in discussion. Such as which enzymes with catalytic activity and transporter activity play the role in metabolism of starch and sucrose. 

Response：Thank you for pointing this out. Using iTRAQ phosphoproteome strategy, we found the high temperature stress mainly effects the starch and sucrose metabolism. Recently, several studies pay attention to sugar acts as signal mollecular to regulate vegetative growth transfer to reproductive stage in plant [1,2]. In this case, we identified some enzymes with catalytic activity implicated in carbohydrate metabolism were up-regulated after the hihg temperature sress, such as, alpha-trehalose-phosphate synthase (catalyzes the production of trehalose from glucose-6-phosphate), glucose-6-phosphate isomerase 1, chloroplastic (Promotes the synthesis of starch in leaves). In line with the variation of sugar component, expression of some carbohydrate transporter were changed, for example, monosaccharide-sensing protein 2 were down-reulated after high temperature stress, resulting the high level of glucose and fructose. In response to high level of sugar concentration, proteins aquaporin PIP2-1-like and dehydrin Rab18-like, which associated with osmoregulation were also up-regulated. 

It is well known that sugar is crucial for basal cell metabolism and plant developmental transition. Rencent studies have showed that variation of sugar concentration can regulate plant growth, high carbon availability can be sensed by trehalose-6-phosphate, and TOR protein kinase, activation of these kinase stimulateds protein translation and promote the plants transition to the generative phase[3,4]. In this study, we found the phosphorylation level of trehalose-6-phosphate, and TOR protein kinase were up-regulated after high temperature stress, this may response to hihg level sugar concentration and resulting in blotting in lettuce. Furthermore, sugar affects cell division and cell cycle, which is essential for plant growth processes. Here, we found high temperature stress mainly effects two key kinase family related to cell division: MAK kinase and casein kinase.

Taken together, the high temperature induced the carbohydrate synthesis enzyme activity, leading to increasing sugar concertration, these high level sugar concentration activate MAK kinase, casein kinase, trehalose-6-phosphate and TOR protein kinase, which play key roles in cell division and growth, then ultimately accelerate the rate of vegetative growth transfer to reproductive growth stage in lettuce, contributing to blotting out. 

We have revised the relevant part of the discussion.

[1] Ciereszko I. Regulatory roles of sugars in plant growth and development. Acta Soc Bot Pol. 2018;87(2):3583. https://doi.org/10.5586/asbp.3583

[2] Astrid W. Transitioning to the Next Phase: The Role of Sugar Signaling throughout the Plant Life Cycle. Plant Physiology, 2018, 176(2):1075.

[3] Schluepmann H, Berke L, Sanchez-Perez GF. Metabolism control over growth:a case for trehalose-6-phosphate in plants. J Exp Bot. 2012;63:3379–3390. https://doi.org/10.1093/jxb/err311

[4] Lee K, Seo PJ. Arabidopsis TOR signaling is essential for sugar-regulated callus

formation. J Integr Plant Biol. 2017;59:742–746. https://doi.org/10.1111/jipb.12560

4. The authors had better add the photo about On day 0, 4, 8, 12, 20, 32 after high temperature treatment and control.

Response：Thank you for pointing this out, we add the photo about on day 0, 4, 8, 12, 20, 32 after high temperature treatment and control in supplement Figure1.

5. The English of your manuscripts must be improved before resubmission. I strongly suggest that you get the assistant from a native English speaker.

Response：We have extensively revised our manuscript, improved the writing style, and we have had the manuscript edited by a professional English language editing service. The proof document was listed below.

---

## [Decision Letter · Decision Letter 1]

7 Dec 2020

Phosphoproteomic Analysis of Lettuce (Lactuca sativa L.) Reveals Starch and Sucrose Metabolism Functions during Bolting Induced by High Temperature

PONE-D-20-23881R1

Dear Dr. qin,

We’re pleased to inform you that your manuscript has been judged scientifically suitable for publication and will be formally accepted for publication once it meets all outstanding technical requirements.

Kind regards,

Yuan Huang

Academic Editor

PLOS ONE

Additional Editor Comments (optional):

Reviewers' comments:

Reviewer's Responses to Questions

**Comments to the Author**

1. If the authors have adequately addressed your comments raised in a previous round of review and you feel that this manuscript is now acceptable for publication, you may indicate that here to bypass the “Comments to the Author” section, enter your conflict of interest statement in the “Confidential to Editor” section, and submit your "Accept" recommendation.

Reviewer #2: All comments have been addressed

2. Is the manuscript technically sound, and do the data support the conclusions?

Reviewer #2: Yes

3. Has the statistical analysis been performed appropriately and rigorously? 

Reviewer #2: Yes

4. Have the authors made all data underlying the findings in their manuscript fully available?

Reviewer #2: Yes

5. Is the manuscript presented in an intelligible fashion and written in standard English?

Reviewer #2: Yes

6. Review Comments to the Author

Reviewer #2: In this study, the sugar and starch play important role during lettuce bolting. The authors had little

discussion about it (line273-277). However more deep discussion is need in this article, such as what’s

the detail function that the sugar and starch play in other species. Or the sugar and starch include

many kinds of chemical compound, what is the function of them in plant development?

The authors response the comments. Please add the response to discussion.

The reference Bavita Asthir's article should be cited in this mansucripts.

7. PLOS authors have the option to publish the peer review history of their article (what does this mean?). If published, this will include your full peer review and any attached files.

Reviewer #2: No

---

## [Editor Report · Acceptance letter]

11 Dec 2020

PONE-D-20-23881R1 

Phosphoproteomic Analysis of Lettuce (*Lactuca sativa* L.) Reveals Starch and Sucrose Metabolism Functions during Bolting Induced by High Temperature 

Dear Dr. qin:

I'm pleased to inform you that your manuscript has been deemed suitable for publication in PLOS ONE. Congratulations! Your manuscript is now with our production department. 

Kind regards, 

on behalf of

Dr. Yuan Huang 

Academic Editor

PLOS ONE